# Methodological Aspects in Randomized Clinical Trials of Nutritional Interventions

**DOI:** 10.3390/nu14122365

**Published:** 2022-06-07

**Authors:** Erika Martínez-López, Edsaúl Emilio Pérez-Guerrero, Nora Magdalena Torres-Carrillo, Andres López-Quintero, Alejandra Betancourt-Núñez, Itzae Adonai Gutiérrez-Hurtado

**Affiliations:** 1Instituto de Nutrigenética y Nutrigenómica Traslacional, Centro Universitario de Ciencias de la Salud, Universidad de Guadalajara, Guadalajara 44340, Mexico; erika.martinez@academicos.udg.mx (E.M.-L.); andres.lopezq@academicos.udg.mx (A.L.-Q.); 2Departamento de Biología Molecular y Genómica, Centro Universitario de Ciencias de la Salud, Universidad de Guadalajara, Guadalajara 44340, Mexico; edsaul.perezg@academicos.udg.mx; 3Departamento de Microbiología y Patología, Centro Universitario de Ciencias de la Salud, Universidad de Guadalajara, Guadalajara 44340, Mexico; nora.torres@academicos.udg.mx; 4Departamento de Disciplinas Filosófico, Metodológico e Instrumentales, Centro Universitario de Ciencias de la Salud, Universidad de Guadalajara, Guadalajara 44340, Mexico; alejandra.bnunez@academicos.udg.mx

**Keywords:** randomized clinical trials, nutritional intervention, methodological design

## Abstract

Nutrition is an essential component when promoting human health. Without a doubt, improving the quality of one’s diet can improve one’s quality of life as a whole and help postpone the onset or control of many chronic diseases. The volume of publications in this field has increased in recent years, in line with increased awareness of the importance of nutrition in health; however, the quality of the evidence on which most nutritional guidelines are based remains low, due to errors in conducting nutritional interventions or because the information is primarily derived from observational studies. To enhance the evidence supporting clinical guidelines in nutrition, the quality of randomized clinical trials (RCT) based on nutritional interventions must be improved; nevertheless, due to their heterogeneous nature and a lack of specific guidelines for designing, performing, documenting, and reporting on this type of intervention, conducting a nutritional intervention is a real challenge. Following a review of the literature on the methodological and ethical standards, as well as four extensions of the CONSORT (Consolidated Standards of Reporting Trials) guidelines that should be considered when implementing a nutritional intervention, seven essential aspects were identified. The current narrative review includes definitions, examples, diagrams, and algorithms regarding aspects of the appropriate study design, the intervention of the control group, the randomization and blinding processes, the study population selection, as well as a description of the type of intervention and the personnel involved in carrying out the study in order to make the implementation of a nutritional intervention easier.

## 1. Introduction

Nutrition is a critical component when promoting human health; it is now regarded as a determining factor in the prevention and treatment of many non-communicable dis-eases. Despite the importance of nutrition in health and the fact that information in this field has increased in volume in recent years, the quality of the evidence on which nutritional guidelines are based is questionable, since the majority of nutrition evidence is based on the findings of observational studies. Furthermore, when the data comes from randomized clinical trials (RCT) based on nutritional interventions, the results are frequently debatable or of little clinical significance due to design limitations [1].

Only 26% of clinical recommendations made by nutrition professionals are currently classified as level I evidence, with the remaining 74% classified as levels II and III. RCTs are an important pillar of evidence in the field of health. Therefore, improving the quality of nutritional interventions is necessary to improve the evidence quality of nutritional recommendations [2,3].

A nutritional intervention is a set of actions designed to change a nutritional aspect in an individual or a population. Nutritional intervention involves a wide range of activities, from nutrient administration to the implementation of a nutritional education program [4]. The World Health Organization (WHO) classifies nutritional interventions into four types:Behavioral interventions, which aim to modify eating habits by changing them.Fortification, which is the addition of nutrients to basic foods.Supplementation, which entails administering a specific nutrient to a specific population.Regulatory interventions, the goal of which is to regulate certain activities in order to modify nutrition and improve health [5].

A RCT, regardless of the type of intervention, is essentially a research design that must meet two requirements: the manipulation of a variable (the independent variable), and the random allocation of the intervention between study groups [6]. In addition to these two fundamental aspects, a set of guidelines has been established that must be considered in RCT reports, with the goal of ensuring the quality of the research and providing sufficient evidence to make well-informed health-care decisions [7,8].

Improperly conducting a RCT can significantly reduce the value of the results and the usefulness of an investigation, resulting in time and money being wasted, in addition to the risks to which the study participants may be exposed. The International Committee of Medical Journal Editors and many peer-reviewed journals have adopted the CONSORT (Consolidated Standards of Reporting Trials) criteria to help improve the quality of RCTs and their reports [9].

The CONSORT guidelines were published for the first time in 1996; initially, they only considered clinical trials based on pharmacological treatments. However, after an analysis of the RCTs published up to that point, it was discovered that non-pharmacological therapies accounted for one out of every four publications in 2000. This highlighted the need for updates to improve the quality of non-pharmacological RCT methodology and final reports [7,8].

In general, four CONSORT extensions can be extremely beneficial in the development of nutritional interventions. “Non-Pharmacologic Treatment Interventions” is a useful addition to most nutritional interventions. If the intervention involves the consumption of an herbal compound or a nutritional supplement containing herbs, the extension “Controlled Trials of Herbal Interventions” should be considered. Furthermore, depending on the chosen design and the research objective, the “Non-Inferiority and Equivalence Trials” extension may be useful, for example, in the case of a parallel design in which the efficacy of one nutritional intervention is compared to another or a pharmacological therapy is compared to a nutritional intervention. Finally, the “Cluster Trials” extension will be useful when the intervention is aimed at groups [7,10,11,12].

Despite updates and extensions to the CONSORT guidelines that may be useful in nutritional interventions, there is no specific guide that takes into account the needs of a RCT based on a nutritional intervention as a whole. The lack of a guideline that incorporates the characteristics that nutritional interventions must meet raises the possibility of bias in the methodology or in the reports, reducing the certainty of the results when they are evaluated using tools such as GRADE (Grading of Recommendations, Assessment, Development, and Evaluations) or equivalents [13].

As a result, the goal of this narrative review is to discuss the critical methodological aspects that must be considered when conducting a RCT based on a nutritional intervention. This is to ensure the validity of the results, taking into account the guidelines that they share with the applicable CONSORT criteria and reports that highlight identified weaknesses in nutritional interventions. In general, it includes the following aspects: a proper study design selection, an intervention of the control group, randomization and blinding processes, study population selection, as well as description of the type of intervention and personnel involved in carrying it out.

## 2. Methods—Literature Search Strategy

The current narrative review was conducted in three steps, the first of which was a literature search using the MEDLINE, Web of Science, and Cochrane databases. Articles published between 2000 and January 2021 were included. The search terms used were: ((Conduct a Randomized Clinical Trials) OR (Trials, Randomized Clinical design) OR (Conduct a Controlled Clinical Trials)) AND ((nutritional interventions design) OR (Nutrition Intervention Trials design) OR (Nutritional support)). Additionally, the following search equations were implemented: ((Randomized Clinical Trials design) OR (Trials, Randomized Clinical design) OR (Controlled Clinical Trials design)) AND ((Study population) OR (study target) OR (Participants) OR (Control group)) AND ((Allocation, Random) OR (Randomization) (Blinding)).

From the results of the initial search, two types of articles were chosen: those that discussed the methodological components of conducting a RCT and those that highlighted major points or flaws in RCT, including nutritional therapies. In the first instance, articles that corresponded to a nutritional intervention were not examined.

Furthermore, useful CONSORT extensions in nutritional interventions were identified, analyzed, and included in the review. “Non-Pharmacologic Treatment Interventions”, “Controlled Trials of Herbal Interventions”, “Non-Inferiority and Equivalence Trials” and “Cluster Trials” were considered.

Following document verification, seven aspects were chosen to be included in the current work, which correspond to the RCT design, control group intervention, blinding, randomization, study population selection, and description of the intervention and the study staff involved in carrying out the intervention. These points were included because they are essential components of a nutritional intervention and they are often not reported or carried out adequately in such interventions.

To make the information presented more user-friendly, examples of RCTs were included for each of the points addressed in the work. For the selection of the examples, a search for nutritional recommendations in clinical practice guidelines for common diseases such as irritable bowel syndrome, hypertension and diabetes was carried out. Additionally, the clinical practice guideline for measles was intentionally included, as it is a re-emerging disease, whose treatment includes nutritional measures. Finally, based on the nutritional recommendations that were considered, a direct search for RCTs related to these recommendations was carried out.

## 3. Randomization

Randomization is a fundamental aspect of RCT; it enables study participants to have the same possibility of receiving any of the treatments, along with preventing people involved in the study, such as researchers and/or volunteers, from being aware of the as-signed treatment the volunteers receive, consequently reducing the possibility of bias and increasing the likelihood of evaluation of the real effects of the intervention [14]. Despite the importance of randomization, it is not carried out or reported in a significant percentage of nutritional interventions. A meta-analysis published in 2017 found that a quarter of the 43 nutrition education interventions included did not perform the randomization process, meaning the results of these investigations are prone to bias [15].

Before carrying out the randomization process, it is important to know that there are different types of randomization; among the most used are simple randomization, block randomization, stratified randomization and covariate adaptive randomization. To select the best type of randomization for a study, one must consider the characteristics of the intervention, the study subjects, and the condition or disease to be studied. The most appropriate types of randomization for nutritional interventions are listed below, alongside recommendations and examples of trials in which they were used. Figure 1 shows a summarized proposal for selecting the type of randomization based on the research characteristics.

### 3.1. Simple Randomization

The simplest randomization technique is equivalent to tossing a coin and, depending on the result, including each participant in an intervention group, regardless of the allocation of previous participants. It is very useful in interventions where numerous participants are included, although, in trials with a few study subjects, it can produce an unequal number of participants between groups, causing a condition known as imbalance, which reduces the study’s power; to avoid such a condition, simple randomization is recommended in interventions with more than 200 study subjects, as the probability of imbalance is negligible with this number [14,16,17]. Aside from the number of participants, it is essential to consider the type of intervention and the effect it will have on environmental, social, and biological factors. If the previously mentioned factors are decisive in the results, simple randomization will not be the best option, because there is a possibility that a large percentage of individuals presenting the variable that modifies the response to the intervention accumulate in a group, leading to results lacking in validity; stratified randomization will be more appropriate in cases where age, gender, disease stage, education, or another factor determines intervention response [9].

### 3.2. Block Randomization

This type of randomization allows for study groups with the same number of participants throughout the investigation. It is very useful for small samples and when the recruitment of volunteers is slow. It essentially consists of dividing the total of the planned sample into blocks; the number of blocks is determined by the work team. However, each block must include a number of subjects that is a multiple of the total number of interventions that will be implemented. For example, if the authors determine that the total size of a sample will be 90 subjects and three interventions are planned, 10 blocks could be carried out, with nine study subjects in each block, which will ensure that the interventions are applied equitably throughout the intervention [14]. An interesting nutritional intervention published in 1992 under the title A randomized trial of low dose folic acid to prevent neural tube defects. The Irish Vitamin Study Group used block randomization; in this research, the effect of folic acid on the prevention of neural tube closure defects was evaluated, three interventions were proposed and 354 were included; although due to the size of the sample simple randomization could have been an option, the women included were recruited between 1981 and 1988, so due to the long time needed to reach sample size, block randomization was more appropriate [18].

### 3.3. Stratified Randomization

In cases where, together with the intervention, study population factors that can influence the response of the dependent variable are known, stratified randomization is the most appropriate option. This type of randomization is similar to randomization by blocks; however, the blocks are made considering the characteristics of the study subjects that can influence the response of the intervention; these characteristics are called covariates, which must be identified by the researcher to generate the randomization blocks. Once the blocks have been made, a simple randomization is performed within each block to assign subjects to one of the study groups [14]. This type of randomization is very useful in clinical trials based on a nutritional intervention, mainly because, in some cases, some factors (e.g., biological, sociocultural, economic, psychological) can generate different responses to the same intervention. In this sense, there are currently projects that seek to identify and standardize factors that can promote heterogeneous responses to nutritional interventions. One example in this regard is the OBEDIS project (OBEsity Diverse Interventions Sharing—focusing on dietary and other interventions), which proposes four categories that should be considered when carrying out an intervention in patients with obesity: environment and context, lifestyle, characteristics of the subject and complications; from these four categories a minimum set of variables is obtained that should be considered in clinical trials on obesity, with the aim of stratifying patients in a consensual manner and predicting with greater certainty the responses of people to an intervention focused on treating obesity [19].

### 3.4. Covariate Adaptive Randomization

This type of randomization will be useful when the sample is small and there are characteristics of the study subjects that can modify the response to the intervention. It is also known as randomization by minimization since its objective is to minimize the differences between groups. To carry out this type of randomization, the researchers determine before the start of randomization which factors should be equally represented in the two study groups. That is, they identify the variables of interest, subsequently consisting of performing a simple randomization, with an agreed percentage of the sample. In this first group the covariant of interest must be identified, to later randomly assign the new participants, considering the covariates observed in the first randomized sample, thereby intentionally generating more homogeneous groups [14,20]. This type of randomization can be of great help in nutritional education interventions since some factors such as age or schooling can influence the results of the intervention. An example of a nutritional intervention that used covariate adaptive randomization was an investigation that aimed to compare the effectiveness of brief nutritional advice versus behavioral dietary advice in increasing fruit and vegetable intake, because a fairly large population was included. Heterogeneous, initially simple randomization was performed with a percentage of the study population and the rest of the study subjects were assigned according to age, ethnic group, and smoking habit, with the aim of minimizing the differences in covariates between the groups [21].

Properly choosing and using randomization is of great importance as it helps eliminate selection bias, prevents results from being affected by covariates, it ensures the validity of statistical tests used to compare treatments.

### 3.5. Online Resources for Randomization

Randomization can be accomplished in a variety of ways. In this paper, the authors describe instances of software or online resources that can be utilized without a license and are free of charge.

#### 3.5.1. Using R for Randomization

R is a free statistical and graphical computer programming language environment accessible for most operating systems [22]. R is a statistical computing language and environment. Vienna, Austria: R Foundation for Statistical Computing. (See https://www.R-project.org/ (accessed on 30 March 2022) for more information). In the Appendix A, there are some examples, including code for randomization using R.

#### 3.5.2. Online Randomization Tools

Randomization can be performed with the help of several online tools. The following are some of them.

A basic interface for performing block randomization may be found at http://www.jerrydallal.com/random/assign.htm (accessed on 1 April 2022). The designations of the groups of interest (for example, Group A and Group B) and the number of individuals per block are all that are required in this resource. There are a total of four blocks that can be added. All that is required is to press “Generate Plan” once the labels and the number of patients per block have been specified.

Another resource is https://www.randomizer.org/ (accessed on 1 April 2022), which allows generation of random numbers in groups.

## 4. Considerations for the Nutrition Intervention

### 4.1. Study Population

The selection criteria for the study population in a RCT must be specific, since it is necessary to obtain a homogeneous sample, which has a high probability of benefiting from the intervention. In addition, these criteria are essential to better know the population to which the results will be applicable and to provide external validity to the research [23,24]. In a nutritional intervention-based-RCT, like any other RCT, it is preponderant to define the study population in detail. In the first instance, it is essential to properly report, as needed, age, sex, socioeconomic and sociocultural characteristics, and health status (e.g., primary disease, stage, type, comorbidities, pharmacological therapies) [23,24].

Regarding the last point (health status), if the disease studied has subtypes, it is essential to specify in which subtype the intervention was applied and how the effect of the intervention was assessed. For example, one of the main challenges in establishing evidence for the therapeutic effect of the low FODMAP (Fermentable Oligosaccharides, Disaccharides, Monosaccharides And Polyols) diet (LFD) in irritable bowel syndrome (IBS) is that most studies do not assess differences in response to the LFD between IBS subgroups; furthermore, no unified assessment scale, such as the IBS-SSS (IBS Severity Scoring System), is used to assess symptom improvement [25].

### 4.2. Participants, Care Providers and Care Settings

In some cases, depending on the type of intervention (for example, in behavioral interventions or behavior modification), it is necessary to describe the characteristics of the care providers (those who carry out the intervention) and, if applicable, the characteristics of the environment where the intervention is carried out. In the first instance, the experience of care providers can influence the results; it is likely that, the more experience the provider has, the better results he can achieve. Thus, it becomes necessary to report, as appropriate, the qualifications of the care provider, the years of practice, the number of interventions carried out and the specific training before the start of the trial, in order to allow the results to be applicable and reproducible. Regarding care centers, it is important to report when the intervention is performed in a hospital setting, for there is a positive association between hospital volume and results [8].

### 4.3. Intervention

One of the fundamental characteristics of an RCT is the intentional modification of the independent variables. In RCTs that use pharmacological therapies, it is easy to identify and manipulate the independent variable. However, in a RCT based on a nutritional intervention, it is sometimes difficult to identify and standardize the elements of the intervention that function as “active ingredients” [7]. In this sense, to carry out an effective intervention, it is necessary to identify the components of the intervention that can modify the dependent variables. Subsequently, the researcher should adapt them to the characteristics of the target population, to finally standardize the intervention itself. In order to achieve this standardization, it is necessary to consider and plan the content of each session, the form of application (individual or group), the strategies for the supervision of the sessions, the identification of information exchange between participants of the different study groups, the validity of the instruments used to provide information, the comprehensive programming of the sessions and the total duration of the intervention. Once standardized, a strategy should be considered to evaluate the adherence of care providers to the protocol, in order to ensure that the intervention is uniform. Moreover, it is necessary to assess the adherence to the intervention; once these considerations are taken into account, it is possible to support the internal validity of the study [7,8].

In case the intervention involved a supplement, it is required to include the name of the substance, registration, dose, time and route of administration; if the supplement is a herbal compound, for example, green tea, it is crucial to follow the CONSORT “Recommendations for reporting randomized controlled trials of herbal interventions”, which specifies how to include the scientific name of the plant, characteristics of the product (e.g., part of the plant used, whether the herb is fresh or dried), dose, concentration of active ingredient, form of standardization, and supplier, among others [12].

Finally, if the intervention’s goal is to evaluate the effect of a specific nutrients, five factors must be considered:Assess nutritional status, that is, the state of the nutrient under study. For this purpose, a valid and reliable biomarker, e.g., serum 25-hydroxyvitamin D concentration, should be identified to determine the vitamin D level. In some cases, where the hypothesis suggests that deficiency of the nutrient under study is associated with disease, levels may be used as an inclusion criterion. However, when effect thresholds are unknown, including participants with a wide range of nutrient levels will allow determination of whether the intervention is more or less effective based on nutrient status [9,26,27,28].Administer adequate doses of the nutrient under study. Using different doses within a plausible range will allow identification of threshold responses to the nutrient [27].Document measurements throughout the intervention, specifying changes in nutritional status [27].The hypothesis should test whether the improvement in nutritional status produces the desired effect [27].Consider other nutrients involved in the response. Because nutrients interact with each other, an organism’s ability to respond to one may depend on the status of another; for example, bone gain following calcium and vitamin D supplementation depends on protein intake. Similarly, it is important to consider whether co-administration with foods, nutrients or bioactive compounds may affect the bioavailability or efficacy of the nutrient under study [9,28].

## 5. Study Groups and Intervention Designs

Since the fundamental objective of a RCT is to test the effectiveness and/or safety of a therapeutic choice, at least two study groups must be included in the research, one that will receive the intervention, and another which will receive a “control” treatment [29]. The design can change according to the characteristics of the intervention. Figure 2 shows an algorithmic diagram showcasing the selection of the most appropriate design. Parallel, crossover, factorial and cluster are the four most used study designs, and they are described as follows.

### 5.1. Parallel Design

Multiple study arms can be included in the parallel design. Each study arm receives a distinct intervention, and study participants are assigned to one of the arms at random. Following the randomization procedure, each participant will remain in his or her assigned treatment arm for the duration of the study. This design is the simplest and allows two or more therapies to be compared simultaneously [30]. An example of this type of design in nutrition was carried out by Hussey and Klein in 1990; through a parallel design RCT, they compared the effect of vitamin A administration against placebo, concluding that vitamin A reduces mortality caused by measles in minors [31].

The use of placebo in the control group is only possible in the parallel design when there is no nutritional or pharmacological therapy to compare it to. When a therapy is already available, it is convenient to compare the new intervention to the existing therapy. In general, when comparing a new nutritional intervention to an existing therapy, equivalence is not sought, for this aspect is primarily applicable in pharmacokinetic studies; in-stead, non-inferiority should be sought, which essentially aims to determine if the new nutritional intervention is not less efficient than the reference treatment [11].

It is critical to understand that, in order to justify an intervention with a new nutritional therapy, the new therapy must have some advantage over the existing intervention, such as greater availability, lower cost, fewer adverse effects, or better acceptance, when conducting a nutritional intervention in a parallel design that compares the new intervention to an existing therapy. For this purpose, reviewing the CONSORT extension “Reporting of Non-inferiority and Equivalence Randomized Trials” is highly recommended [11].

### 5.2. Crossover Design

When studying chronic diseases that are stable over time, and whose non-permanent intervention effects may be noticeable in the short term, it is possible to use a crossover design. This allows for greater biological homogeneity, since all groups receive different interventions at different times, so the groups can be compared against each other at the end of the study, thus contributing to confirming the results of the intervention. Besides, more than two groups can be included and the incorporation of participants or groups can be sequential [32,33].

A critical aspect to take into account when conducting a crossover trial is the “washout” period. This period is between the end of the first intervention and before the start of the second intervention, that is, when the treatment is crossed over; the aim of this period is to avoid a possible “carry-over effect”, meaning that the residual effect of the first treatment administered does not influence the evaluations of the second treatment. In cases where the half-life of the administered product can be measured (e.g., a food supplement), it is generally recommended that the washout period should be at least five times the half-life of the treatment under study [34].

A clear example of this design was the DASH diet trial (Dietary Approaches to Stop Hypertension), in which the study groups received both the control diet or standard diet and also the DASH diet at two different times, to evaluate its effectiveness in reducing blood pressure [35]. In general, when deciding on a crossover design, two factors must be considered. The first is that the disease or condition being studied must be chronic and stable, and the characteristics must not change between the two study periods. The second criterion that must be met is that the intervention’s effects must not be irreversible, and changes should preferably be visible in a short period of time [30]. In the case of the DASH diet, both criteria are met; hypertension is a chronic and stable disease, and the intervention does not result in permanent changes, that is, a patient with hypertension can lower his blood pressure while adhering to a specific nutritional plan, but his blood pressure may rise again after quitting or adopting harmful habits.

It is worth mentioning that the crossover design is unsuitable for educational nutritional interventions, because the information gained during the intervention can be permanent or on a long-term basis [36]; it will also be ineffective in diseases that vary during the intervention. For example, administering vitamin E to patients with non-alcoholic steatohepatitis can result in a decrease in hepatic steatosis and lobular inflammation, clinical or histological improvement complicates the comparison of the groups at the point of crossing the interventions [37].

### 5.3. Factorial Design

This is used to test more than one intervention in the context of a single study; that is, when the effect of one intervention is thought to be dependent on the presence or absence of another intervention, a factorial design is preferable over an independent parallel design [38]. In this design, the 2 × 2 approach is the most common. With this approach four research groups are generated; the first receives treatment one + control one, the second treatment two + control two, the third group treatment one + treatment two and the fourth group control one + control two. With this type of design, it is possible to examine the additive, synergistic or antagonistic effects of the therapies to be investigated. Yet, given that more study groups and interventions are required, these designs generally need a larger sample size and are more expensive [9]. To better understand this type of design, this review will use Kukuljan’s 2011 research as an example. The study looked at the synergistic effect of physical exercise and calcium and vitamin D consumption on the structure and mineral density of bones in men aged 50 to 79 years old. Four groups were formed: the first received vitamin D and calcium, the second only exercise, the third calcium, vitamin D, and exercise, and the fourth received no intervention [39]. In this example, in a logical sense and based on previous evidence, two potentially synergistic interventions are carried out in the same investigation, for which the factorial design was very appropriate.

Nutritional interventions can benefit from using the factorial design because it is possible to combine different types of nutritional interventions to achieve better results, such as a supplementation with a dietary pattern. For example, it has been described that a LFD is associated with improved symptoms of IBS, while peppermint oil has been proposed as a safe and effective therapy for pain and global symptoms in adults with the same disease [25,40]. Considering two separate approaches appear to work for the same illness in this situation, they may easily be combined into one to provide more comprehensive therapeutic alternatives.

### 5.4. Cluster Design

When an intervention targets natural groupings of people, the cluster design is the most appropriate. In this type of design, interventions are assigned to groups, so they are carried out in settings where it would be difficult to assign different treatments to participants within the same group (e.g., nursing homes or a school canteen). It is important to consider that, in this type of design, it is essential to plan in detail the number of clusters and the number of individuals within a cluster [9].

In this type of design, randomization of the intervention is carried out at the cluster level. Clusters are usually randomized all at once rather than one at a time, as in individual randomized trials. Additionally, before conducting a cluster trial, it is important to consider that trials with only one intervention group per arm should be avoided, as there is a greater likelihood of confounding the effect of the intervention. According to CONSORT guidelines it has been suggested that the minimum number of clusters per arm for a valid analysis should be four [10].

An example of a cluster design was the SMART protocol (Small, Measurable and Achievable dietary changes by Reducing fat, sugar and salt consumption and Trying different fruits and vegetables), which compared the efficacy of educational intervention based on information technology against traditional nutritional education, with the aim of identifying which of the strategies improves eating habits to a greater extent [41]. The cluster design will be especially useful in nutritional educational interventions, as these are typically aimed at groups, such as students in a classroom or relatives in families. However, because cluster randomized controlled trials require special considerations for analysis, interpretation and reporting of results, we suggest reviewing CONSORT’s “Extension to cluster randomized trials” to be clear about the aspects that should be reported in interventions under this design [10].

## 6. Blinding

Blinding aims to avoid the expectations of both participants and trial investigators, which might affect the actual results of the intervention. There are different models of blinding, such as single-blinding, in which the volunteers do not know the intervention they are receiving, double-blinding, in which the volunteers and trial investigators do not know the assigned interventions, and triple-blinding, in which, in addition to the volunteers and researchers, the people analyzing the data are unaware of the intervention. The blinding process can be made much easier using a drug or nutritional supplement, but when dealing with food it can be more difficult [3].

In the case of study subjects, not only must the group to which they belong be blinded, but the study hypothesis must also be blinded. The blinding of the study subjects with respect to the group to which they belong (control or intervention) is important because, if the participants have some favorable belief in the intervention, it can influence the results. For example, an investigation published in the year 2019, entitled Studying a Possible Placebo Effect of an Imaginary Low-Calorie Diet presents a randomized clinical trial in which obese subjects underwent a similar regimen of physical exercise and an isocaloric nutritional plan; the control group was informed that they were on a balanced diet, while the intervention group was told that they were under a hypocaloric diet with a deficit of 5500 kcal per week. It was found that the intervention group obtained better results, and it is concluded that “the opportunity of being allowed to participate in an experiment, which is supervised and controlled by professional dietitians and strength-training coaches, could have been a great stimulus for some participants to reduce their calorie intake and/or energy expenditure further than prescribed and lose weight as a result”. Similarly, it is important to blind the hypothesis, since if the participants know it and realize they are not in the intervention group, this can influence the results [42,43,44].

To improve the quality of evidence for nutritional recommendations, the blinding process must be optimized. A clear example is the effect of LFD on IBS; although LFD has been recommended for the treatment of IBS, the evidence supporting this recommendation is of low quality [45]. In this regard, a review published in 2021 that assessed the evidence using the Grading of Recommendations Assessment, Development and Evaluation (GRADE) system found that inadequate blinding is one of the reasons for the low level of quality of the evidence supporting the LFD recommendation [25]. However, the blinding process is difficult because most IBS patients are already familiar with the LFD.

Staudacher et al. used a placebo (simulated) diet to solve this problem; according to their proposal, the simulated diet should be of equal difficulty, restrict a similar number of staple and non-staple foods, the duration and intensity of the counselling should be similar and should not affect other nutritional components of the diet besides FODMAPs (or components of the diet under investigation) [46]. In case blinding to the intervention group to which they belong cannot be ensured, at least the participants of the study hypothesis should be blinded [42,43].

Blinding of those who implement the intervention is also recommended; however, when it is not possible due to the nature of the research, it is recommended to blind to the study hypothesis, the inclusion criteria, and the results of interest. To do this, one of the actions to be implemented is to ask the participants not to mention their results to those who implement the intervention. In addition, it will be important to maintain standardization in the intervention using manuals and to evaluate fidelity of the intervention protocol. Regarding the outcome assessor, blinding of the intervention groups should be sought. In other words, these people should not know which intervention groups the participants belong to, especially when it is not possible to blind the participants of the assigned treatment. This blinding can be done by asking participants not to tell the outcome assessor what treatment they receive. It is important to explain to participants, in non-scientific language, why blinding is important and how they can help maintain it. If the outcome assessor is aware of the intervention groups, it is suggested that he or she does not get involved in treatment administration and in the participants’ recruitment process. It will also be necessary for the main result of the study to be objective and/or to blind the purpose of the study and/or the hypothesis in question to the outcome assessor. A lack of blinding at this level may result in assessment bias and lead to inflation or lack of significance in the treatment effect. To maintain blinding, it might work to separate those giving the intervention from the outcome assessors [42,43].

Ideally, the principal investigator should be blinded to the results and treatment assignment and he or she should have the minimal contact possible with the participants. The primary hypothesis researcher cannot be blinded because he or she designed the study. The involvement of the principal investigator in the implementation of the treatment or evaluation of the result can introduce bias in the study, since they tend to have a great interest in the development of the project. Once the data collection and analysis are complete, the principal investigator accesses the information to write the main results’ scientific papers. This blinding is used in both pharmacological and non-pharmacological studies to minimize bias in the final analysis [42,43].

Finally, the statistician will need to be blinded to the intervention groups. One way to go blind is to give each study group in the database a name that has nothing to do with the intervention (for example, “group one”, “group two”, and “group three “). This blinding allows the statistician to prepare statistical analyses “a priori” and avoid performing extra analyses, instead focusing on finding a statistical test that yields a significant result in favor of the intervention of interest. It is also critical that the data analysis is performed by someone who is not involved in the intervention implementation or evaluation. Other factors to consider in blinding regard the documents and processes, such as the informed consent letter. It is important to avoid disclosing which intervention is superior. In addition, neither the study hypothesis nor the scientific distinctions between the intervention groups are communicated. Also, if an abstract for the study protocol is released, it should be worded in a general manner so that participants who read it do not have specific expectations about the main intervention [42,43].

## 7. Intervention in the Control Group

It is often the case that, when carrying out an intervention, the following question arises: is it strictly possible not to carry out any intervention or to give a placebo to the control group? The specific answer turns out to be yes. In fact, the treatment of the control group can be a placebo, no intervention, usual care, active treatment or a waiting list; however, the choice depends on the nature of the intervention and the available therapies [8]. When there is a proven intervention for the condition under investigation, it is not appropriate to use a placebo or a non-intervention approach, due to ethical reasons and the very scientific value of the research.

In the first instance, standing in accord with the provisions of the Declaration of Helsinki of the WMA (World Medical Association) ethical principles for medical research in human beings, “although the main objective of medical research is to generate new knowledge, this objective should never take precedence over the rights and interests of the person who participates in the research”. The use of a placebo and non-intervention approaches are only acceptable when there is no effective intervention to treat the condition being investigated [47]. Additionally, from the point of view of scientific and social value, if there is an effective therapy to treat a particular condition, it is not appropriate to use placebo or no therapy whatsoever. RCT must be carried out under the hypothesis that the new therapy is superior to conventional therapy, as there is no ethical justification for conducting a RCT with therapeutic options that have no advantage over standard approaches. In some cases, it is possible to carry out a non-inferiority trial, which is used to prove that a new intervention is not less effective than standard therapy, and it presents an advantage in certain critical aspects such as availability or cost [48].

### 7.1. Types of Interventions in the Control Group

Ideally, the intervention should modify the dependent variable due to its pharmacological, toxicological and/or nutritional properties. Nonetheless, in some cases, the effects of the intervention may be the result of the expectations of the study volunteers (placebo effect), hence the importance of the control group, to be certain of the intervention effect [3].

In addition to the effect observed by the expectations of the volunteers, personal and cultural beliefs regarding diet choice, sensory satisfaction, taste preferences and the sup-port of a nutritionist, among other variables, can influence the results of clinical trials based on nutritional interventions [3]. Figure 3 shows a proposal for the selection of the intervention in the control group, and some possibilities for this group are described below:

#### 7.1.1. No Intervention

It is possible not to apply any intervention when there is no established therapy for the study condition being studied. It is important to note that the absence of any intervention does not imply the control group receives no medical care; instead, it simply means that, as part of the research, one potentially functional intervention will not be compared to another. For example, in 2008 Neal et al. published a clinical trial assessing the efficacy of a ketogenic diet in lowering the frequency of seizures in children who had not responded to antiepileptic drug treatment. The only intervention in this study was the implementation of a ketogenic diet to 73 out of the 145 minors involved; there were no changes in any of the participants’ medical assistance during the study [49]. In the previous example, the lack of a standard treatment against which to compare the ketogenic diet means that there is no acceptable intervention for the control group. Furthermore, given the medical support measures that the patients received were not a reason for comparison with the ketogenic diet, they were not required to be removed. When no intervention is administered to the control group, there is naturally no treatment to blind, that is, the participants are aware of the group to which they belong, which can increase the expectation of a possible benefit in the intervention group and an expectation of lack of benefit in the control group. This could generate a possible bias in treatment trials with subjective results such as in cases of evaluating the sensation of appetite, satiety, quality of life, and reported symptoms, among others. For this reason, in the context of interventions that assess subjective outcomes, researchers should be highly cautious when applying a non-intervention scheme in the control group [3].

#### 7.1.2. Waiting List

The control group remains on a waiting list before receiving the intervention. That is, a design similar to the crossed one is used, in which two groups are created, one control and one intervention; initially, the control group will not receive any intervention; during this period, the effect of the intervention can be compared with the non-intervention; later, after the comparison period, the control group will receive the intervention, which will serve to evaluate its effect on the control group [3]. In the first instance, the waiting list control has an ethical advantage because all study subjects will receive the intervention; however, there is concern that using this type of control may overestimate the effects of the intervention, particularly in research whose goal is to modify behavior; in this sense, it has been described that the controls that are included in a waiting list “perceive that they are expected to ‘wait’ to change until they receive the intervention”, while in the designs that do not use a waiting list, the participants of the control group tend to improve [50]. However, contrary results have been reported, such as studies in which subjects on waiting lists for weight loss protocols lose weight while on the waiting list, implying that, depending on the nature of the intervention and the condition being studied, the use of waiting list control may overestimate or underestimate the intervention’s effects [3]. The waiting list control is considered a control without treatment, so it will be very useful when studying conditions for which no established treatment or therapy exists; it will also be very useful when implementing a novel strategy that lacks a point of comparison, such as an investigation led by Payal Agarwal in 2019, which looked at the effectiveness of using a mobile application to help patients with type 2 diabetes control their glucose levels [51]. The waiting list control is acceptable in the previous example because the intervention (mobile application) does not attempt to replace or be compared to the established therapy for the patients, but rather serves as an additional tool to the therapy that helps motivate, raise awareness, and educate patients improve their glucose control. Therefore, the usage of the waiting list is adequate because there is no alternative instrument or strategy that pursues the same goals and thus with which it can be compared, as well as the fact that the use or absence of use of the mobile application does not represent a threat to the study subject.

#### 7.1.3. Active Control or Standard Treatment

This approach basically consists of comparing a standard treatment or giving dietary advice that is known to have some established efficacy against a novel nutritional intervention. It is often used to establish the effect of a new dietary intervention as equivalent or superior to current practice, plus all participants receive a treatment, making it more ethically acceptable. Furthermore, in the case of pharmacological therapies, patients prefer to participate in active comparison trials rather than placebo-controlled trials when evaluating the efficacy of the drug. Compared with waiting list or no intervention in the control group, active treatment or standard control is less likely to be biased. However, when the control receives ‘standard advice’, which has been customary in practice clinics for some time, it is likely to generate an unequal distribution of expectations, which could reduce the internal validity of the trial [3]. From an ethical standpoint, this design is very acceptable because no study subject is left without a treatment option; this type of control will also be very useful when a bioequivalence or non-inferiority trial is conducted. However, before attempting any new intervention, it is critical to have scientific evidence to support its efficacy and safety. The use of active control or standard treatment has been widely used in nutritional interventions because it is possible to compare the effect of a nutrient against a drug. An example of an active control is a 2013 study that compared the effect of metformin versus chromium picolinate to treat insulin resistance in patients diagnosed with polycystic ovary syndrome resistant to clomiphene citrate [52]. In this case, the active control was the best option, because when proposing chromium picolinate as a therapy option, it is necessary to compare it with the best available therapy. When proposing a new therapy, such proposal is expected to surpass the existing options considered as gold standard. Furthermore, for ethical reasons, if there is a therapy for the study condition, it is not convenient to discontinue administering it to the study subjects.

#### 7.1.4. Placebo

This consists of administering an inert substance, as similar as possible, to the intervention substance of interest. It is considerably easy to carry out a placebo intervention when it comes to a drug or a supplement. Conversely, when it comes to dietary interventions or dietary advice it is significantly more complicated. Despite this, it is possible to carry out the use of placebo in nutritional interventions, initially by identifying the “active components” of the diet of interest, which will later serve to generate a placebo diet. It consists of a nutritionally compatible menu in all aspects, except in the “active component” that is investigated, allowing both diets (control and intervention group) to be almost indistinguishable [3]. An intervention directed by Barrett published in 2010 serves to illustrate how one can proceed with a placebo based on a complete diet; the goal of the research was to evaluate the effect of a FODMAP-rich diet on the amount of water and fermentable substrates excreted in the ileal effluent of subjects who underwent ileostomy [53]. To create a placebo diet, the first step was to identify the part of the diet that works as a “active ingredient”, which in the case of the mentioned intervention is FODMAP, so the placebo diet was free of these, but the rest was similar in composition, that is to say equivalent in energy, macronutrients, and fiber, so the two study groups received a similar diet that only differed in the compounds considered as the intervention.

## 8. Summary of Major Methodological Aspects

Some of the summarized methodological aspects apply not only to those based on nutritional interventions. However, it is noteworthy that the type of intervention (behavioral, fortification, supplementation, or regulatory) and the intervention in the control group (no intervention, waiting list, active control or standard treatment, placebo) are crucial for the success of the trial. The major aspects to be considered when performing RCTs based on nutritional interventions are summarized in Figure 4.

## 9. Conclusions

Carrying out a RCT based on a nutritional intervention is a real challenge; still, generating quality knowledge and recommendations based on high-level evidence is a patent commitment of the scientific community towards the general population. Currently, the information available to guide the researcher on this particular aspect is scarce. Even the CONSORT declaration, despite being an excellent resource, does not contain a specific description to present the information of a RCT based on a nutritional intervention. Figure 4 depicts the points to follow when carrying out a nutritional intervention, thus summarizing the aspects discussed throughout this document.

Although the present review can function as a guide, constant updating is necessary, in accordance with the progress of basic sciences and ethics, which keeps the methodology on par, in order to generate information of greater scientific quality.

Seven methodological aspects are suggested as essential for the success of a RCT based on nutritional intervention with the following recommendations:Study population: Specify selection criteria, define in detail the population to study and select a recruitment method; assess the external validity of the trial.Participants care providers and care settings: The participants in the trial should be selected according to their qualifications, years of practice, number of previous interventions carried out and special training before the intervention.Intervention: Identify the components of the intervention that can modify the dependent variables and standardize the intervention as appropriate for behavioral, fortification, supplementation or regulatory interventions. Adherence of care providers to protocol is important as the participant adherence to intervention; provide mechanisms for internal validity of the study.Study groups and intervention designs: Determine if the intervention will be carried out in groups or individually; select a study design in accordance with the characteristics of intervention, such as parallel, crossover, factorial or cluster design.Intervention in the control group: Analyze if there exist a gold standard for the disease/condition under investigation and select the best option: no intervention, waiting list, active control/standard treatment or placebo.Randomization: Consider population size, number of study subjects in each group and recruitment time.Blinding: To avoid expectations interference in the results, choosing single-double or triple-blinding is the best option. If this is not possible, it is suggested to blind the hypothesis and/or results.

## Figures and Tables

**Figure 1 nutrients-14-02365-f001:**
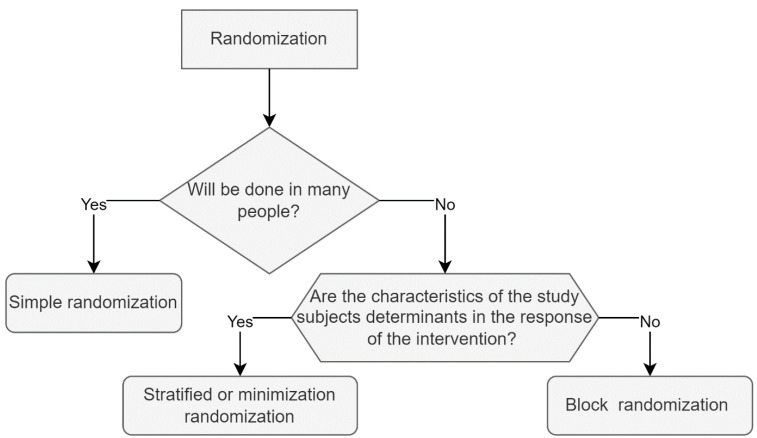
Randomization: proposal to choose the most appropriate type of randomization. In this proposal, it is recommended that the authors ask themselves three basic questions to determine which sort of randomization is best appropriate for the study. The first is whether the study population is large. If “yes”, simple randomization is the best solution; the second question is: is the sample small, is the recruitment time long, or do the authors need the same number of study subjects in both groups? Block randomization will be the best option if the answer is yes. Third, are the covariates important for the intervention? If that is the case, stratified randomization or minimization are the best options.

**Figure 2 nutrients-14-02365-f002:**
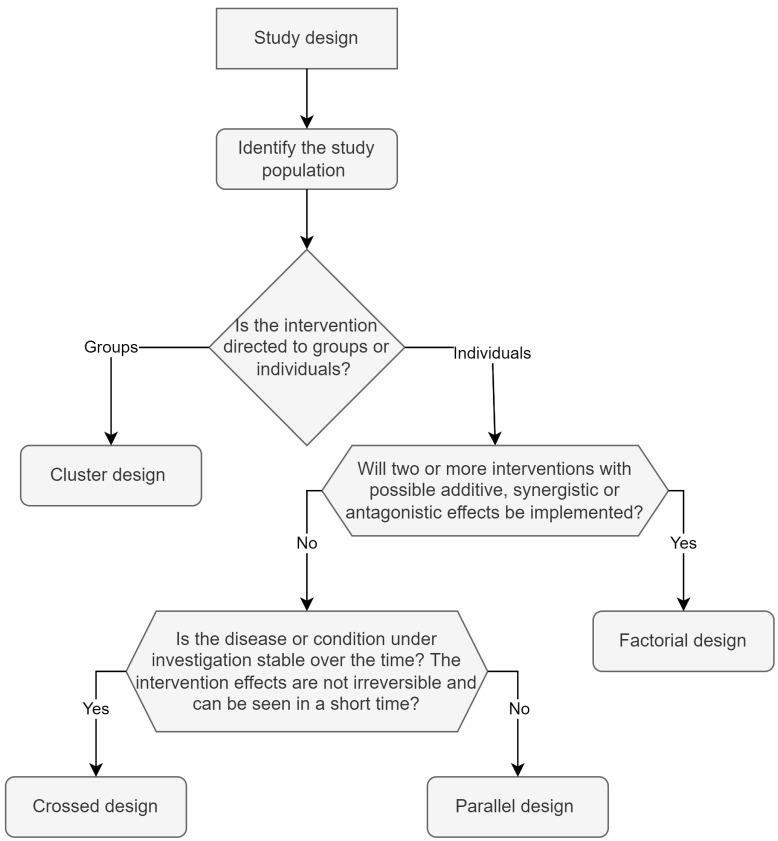
Study design: proposal for the selection of the most appropriate design. To choose the study design, it is recommended to first determine whether the intervention will be carried out in groups or individually. In case the intervention cannot be carried out in individuals, the recommendation will lean toward the cluster design. If more than two interdependent interventions will be used in the study, the factorial design is the best choice. The crossover design will be very appropriate if a stable disease is studied over time wherein effects of the intervention are observed in a short time and are not of a permanent nature. The parallel design will allow researchers to compare two or more unrelated interventions in diseases that can be exacerbated or cured by the intervention.

**Figure 3 nutrients-14-02365-f003:**
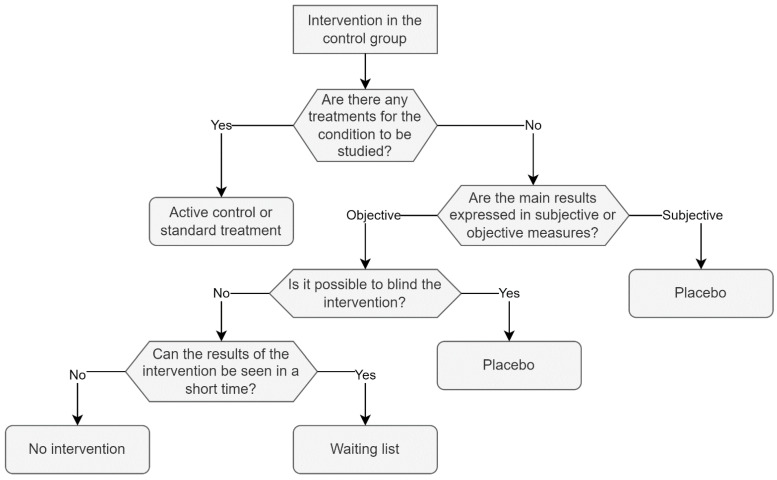
Intervention in the control group: proposal for the selection of the intervention in the control group. If there exists a therapy for the disease under investigation, it is suggested to compare the intervention to the available therapy as the best alternative. If there is no therapy or the intervention is not comparable, and if the intervention can be blinded, it is recommended to use a placebo, whether the findings are subjective or objective. Finally, if blinding the intervention is not possible and the outcomes can be seen quickly, the authors recommend a waiting list.

**Figure 4 nutrients-14-02365-f004:**
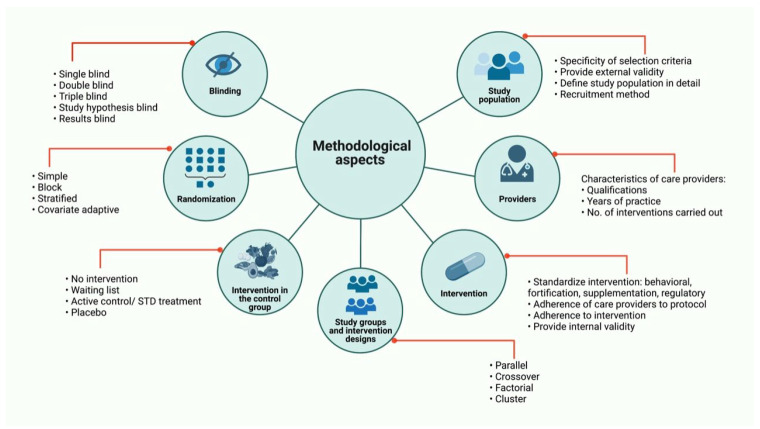
Randomized Clinical Trials based on nutritional interventions. Major aspects to be consider in RCTs based on nutritional interventions are: study population, care providers, intervention, study groups and intervention designs, intervention in the control group, randomization and blinding. Some of these methodological aspects apply not only to those based on nutritional interventions. RCT: Randomized Clinical Trials; STD treatment: Standard treatment; No.: Number.

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
