# Peer review of "Methodological Aspects in Randomized Clinical Trials of Nutritional Interventions"

_nutrients, 2022, doi:10.3390/nu14122365_

Round 1

Reviewer 1 Report

Authors have written a review entitled “Methodological Aspects to Be Considered in Randomized Clinical Trials Based on Nutritional Interventions”. This is a well written review. Please find some minor comments below for improvement.

Should be “a” RCT and not “an” RCT

Line 630: f in figure should be in caps

Reviewer 2 Report

 Title suggestion: Methodological Aspects in Randomized Clinical Trials of Nutritional Interventions 

Please clarify this long run on sentence. The later statement on convenience sampling doesn’t make sense either. Papers better to remove as the main points about the study population has been above prior to this point.

In 73 a RCT based on a nutritional intervention, like any other RCT, it is preponderant to define 74 the study population in detail, in the first instance, it is essential to report, as appropriate, 75 age, sex, socioeconomic and sociocultural characteristics, health status (disease, stage, 76 type, comorbidities, pharmacological therapies, etc.) and the recruitment method, which 77 generally, the selection of volunteers in RCT, is carried out through non-probabilistic, con-78 secutive convenience sampling [9,10].

Section 4.2

The information contained in this section should be more carefully stated. First, it is always good and required to describe the intervention and the administration of intervention in any study. Although the outcomes overall clearly is dependent on the quality and type of intervention, it does not affect the assessment of treatment effect. For example, suppose that intervention is at the patient/subject level among a mixed level of experienced providers (or low to high volume providers). Would this affect the estimation of treatment effect? NO. Randomization would eliminate all such systematic differences that still provide unbiased estimate of treatment effect (treatment difference). How about the case of cluster randomize trials where the intervention is that the provider-level? Certainly if there are few providers in such a design, then it may pose a problem with assessing treatment outcome, but one would presumably not engage in such a study without defining a standard of intervention and ensuring such some standardization of intervention administration. 

The statement below needs to be modified.

… the intervention and another, which will receive a "control" treatment [12]. According to 119 …

Need not be a “control”. It would be two nutritional interventions; it could be one compared to a non-nutritional intervention such as a specific diet regiment vs. a standard of care, or vs. a pharmacologic agent. 

5.1 Need not be two groups.

5.1. Parallel design: in this design 

Capitalize “In”.

5.2

… so the groups can be com-131 pared against each other at the end of the study and provide greater certainty of the effect 132 of the intervention. … 

This statement is not true. There’s not greater certainty on effect of intervention. 

Cross-over design is more efficient requiring half the number of subjects than a comparable parallel design and therefore more cost-effective. However, at the same time, dropouts (equivalent to parallel design) can more substantially biased estimation of treatment effects as there the number of patients is small in the cross-over trial. Further, proper administration of cross-over design should therefore incorporate intermediate measurements, not just baseline and after treatment to reduce potential effects of dropouts. The washout period should also be mentioned as an important aspect of cross-over design. 

Consider an inclusion of parallel and cross-over design diagram; these figures are fairly standard.

There appears to be a misunderstanding of factorial design so Figure 1 is can be confusing. As this paper is a very basic introduction, I suggest removing the factorial design. One can have many treatment groups in both parallel and cross-over design. The authors focus on explaining that carefully. 

At the same time, the clusterd design can also be parallel or cross-over (e.g., hospital or clinic can cross-over).

Therefore, in an introductory paper, you should focus on explaining parallel and cross-over designs, at the individual level and if that the intervention is at the group e.g. provider level then it is called a cluster randomized design and a clustered-randomized cross-over trial.

Sections 6.1.1 and 6.1.2 do not make sense.

There appears to be confusion with blinding. For example, why would there exist a “control” waitlist in a randomized control trial? Subjects are randomized.

6.1.3 Compared 219 with waiting list or no intervention in the control group, active treatment or standard 220 control is less likely to be biased. however,  …

This statement is not true generally. The statement should be attribute to a specific aspect of a study. For example, this is not true in blinded trials of a supplement vs. a control placebo for instance,

Also, grammatical error and throughout. Please proof read carefully.

Section 9 and other places in the abstract and introduction:

I really do not agree with the statement that CONSORT standards do not accommodate methodological aspects of nutritional intervention trial. It clearly is in my view. It is better focus on truly unique aspects of nutritional intervention, which is where the contribution of this paper is. There is nothing new with respect to methodology of clinical trials that would warrant such statement or conclusion.

Round 2

Reviewer 2 Report

Revision okay.

Author Response

We appreciate the time to review this paper with your expert opinion.

This manuscript is a resubmission of an earlier submission. The following is a list of the peer review reports and author responses from that submission.

Round 1

Reviewer 1 Report

The authors summarized some aspects of clinical trials studying nutritional interventions. This is in principal a good idea, however the authors describe many aspects too superficial. The authors would need to go into much more detail and prepare useful suggestions for nutrition studies. 

As an example: The paragraph on placebo only describes what a placebo is in principal but does not give any suggestions how to make a placebo in nutritional trials.

The figure on randomization is nice but has no connection to nutritional studies.

The authors often write "for example" and only give some arbitrary examples - in such a review they should try to compile as much evidence as possible on best practice examples to make such a review useful for readers as a reference source. 

These are only a few examples, the work has to be thoroughly revised in every aspect!

Reviewer 2 Report

The authors have written a review article entitled “Methodological Aspects to Be Considered in Randomized Clinical Trials Based on Nutritional Interventions”. This is a very interesting topic but the manuscript needs to be improved. The authors have presented more of a descriptive summary of the topic rather than a critical discussion. Suggestion is that authors rework the manuscript and ensure that they engage with more than one literature source for sections. Further it would be good to see what other studies have found regarding RCT. A methodology section indicating how you did your data search would add value.

General comments

The introduction can be strengthened to emphasize the importance of this review.

Methodology: It would be useful to indicate to the reader how you went about conducting your literature search (key words, websites, etc) as well as how you filtered the data. What year search did you use?

Present the study design as soon as it is mentioned.

Same comment applies for all figures.

For certain sections there is only one reference used. It is important to go through several pieces of relevant literature when writing a review.

The quality of your figures is not good.

Conclusion: It is good to add the way forward.

Round 2

Reviewer 1 Report

The authors took up my comments and improved the manuscript. The manuscript has no major flaws but it still lacks novelty.

Reviewer 2 Report

Thank you for giving me the opportunity to review the revised manuscript entitled “Methodological Aspects to Be Considered in Randomized Clinical Trials Based on Nutritional Interventions”. This was a much better read. The narrative review flows better and is an interesting read. To further improve the manuscript, please incorporate the comments/suggestions below:  

Introduction

You should start your introduction by explaining to the reader what RCTs are and then move into the history.

Lines 43-53: This is a too long sentence. Please split into a few sentences. Rule of thumb is usually 3 lines max for a sentence.

Line 53: Instead of saying the goal of this document rather say that the goal this narrative review.

You mention in your responses that this is a narrative review thus should be emphasized.

Lines 53-55: “Therefore, the goal of this document is to provide supporting material for better implement a nutritional intervention in relation to the aforementioned aspects.” This sentence does not make sense.

Suggestion: Thus, the goal of this narrative review is to provide insight on various aspects to consider when conducting RCTs so that nutritional interventions can be better implemented. You need to link you closing introduction statement with your title and what is presented in your narrative review.

Authors have improved the introduction. However, it still lacks strong justification on the need for this narrative. Are RCT routinely done? If not why not? Is there a problem with the way it is conducted? Is there a problem with sample size etc. Are these problems related to no standard guidelines for non-pharmacological therapy RCT. Did this result in the need to conduct this narrative review? (To gather information on aspects that one should consider when conducting RCTs to improve nutrition interventions). Authors need to work on this section a little more to strengthen this review. Once this part is revised then authors should revise the abstract to strengthen it.              

  1. Designing and conducting a RCT

What are the set of recommendations? It would add value to explain this to the reader.

Lines 67-72 is not referenced. Further this sentence is really vague. What aspects could influence the validity? You need to be specific.

Sections 2 and 3 should form part of your introduction and not have separate headings. You can start by explaining RCTs and then lead into nutritional intervention and close by giving the importance of conducting this study. You next heading after the introduction should be the Methodology section.

  1. Methodological aspects to consider in a RCT based on a nutritional intervention

Authors have added a summary of the process followed in the introduction section. This has been acknowledged.

  1. Study groups and intervention designs

Suggestion: Please move Figure 1 to page 4. So that the reader can look at it as soon as it is mentioned.

Lines 138-139: instead of saying “The most used are listed below”. Rather say parallel design, crossover design, factorial design and cluster design are the four most used study designs.

Suggestion: Give intro ending with the sentence above. Then have your figure and then write up on all the designs.

Line 144: Who is GD? Write in full.

Line 145: The initial M should be removed from the start of the sentence

Line 152: remove the word “the” before the word “different”

Lines 149-154: consider splitting into two sentences so your message comes across better.

This section has been improved.

  1. Intervention in the control group

Figure 2 should appear earlier on so that the reader can visualize it and thereafter read up on the other points.

This section has been improved significantly.

  1. Randomization

Same as previous comment regarding the figure.

This section has been improved significantly.

  1. Blinding

Additions to this section have improved it.

  1. Conclusions

You should not refer to a figure in your conclusion. The purpose of the conclusion is to sum up the main points and not introduce any new information. However, the figure does add value to your review and should be included earlier on in the review. You can write the points in the narrative in the conclusion.

General comments

  • Numbers under 10 should be written in full.
  • In scientific writing we do not use the word “we”, rather use the authors or this review shows.
  • It would be beneficial to check your grammar
  • Try not to use the word etc. in your narrative review
  • Some formatting is not done as per journal guidelines, please check this.
  • Good range of references.
